# The (Un)Scalability of Informed Heuristic Function Estimation in NP-Hard Search Problems

**Sumedh Pendurkar**                                    *sumedhpendurkar@tamu.edu*
*Department of Computer Science and Engineering*
*Texas A&M University*

**Taoan Huang**                                         *taoanhua@usc.edu*
*Department of Computer Science*
*University of Southern California*

**Brendan Juba**                                        *bjuba@wustl.edu*
*Department of Computer Science and Engineering*
*Washington University in St. Louis*

**Jiapeng Zhang**                                       *jiapengz@usc.edu*
*Department of Computer Science*
*University of Southern California*

**Sven Koenig**                                         *skoenig@usc.edu*
*Department of Computer Science*
*University of Southern California*

**Guni Sharon**                                         *guni@tamu.edu*
*Department of Computer Science and Engineering*
*Texas A&M University*

**Reviewed on OpenReview:** *https://openreview.net/forum?id=JllRdycmLk*

## Abstract

The A* algorithm is commonly used to solve NP-hard combinatorial optimization problems. When provided with a completely informed heuristic function, A* can solve such problems in time complexity that is polynomial in the solution cost and branching factor. In light of this fact, we examine a line of recent publications that propose fitting deep neural networks to the completely informed heuristic function. We assert that these works suffer from inherent scalability limitations since — under the assumption of NP $\not\subseteq$ P/poly — such approaches result in either (a) network sizes that scale super-polynomially in the instance sizes or (b) the accuracy of the fitted deep neural networks scales inversely with the instance sizes. Complementing our theoretical claims, we provide experimental results for three representative NP-hard search problems. The results suggest that fitting deep neural networks to informed heuristic functions requires network sizes that grow quickly with the problem instance size. We conclude by suggesting that the research community should focus on scalable methods for integrating heuristic search with machine learning, as opposed to methods relying on informed heuristic estimation.

## 1 Introduction

Principal computational problems such as planning and scheduling (Wilkins, 2014), routing (Toth & Vigo, 2002), and combinatorial optimization (Papadimitriou & Steiglitz, 1998) are known to be NP-Hard (NP-H) in their general form. Consequently, there are no known polynomial-time algorithms for solving them.

Moreover, for many of these NP-H problems (belonging to the NP-Complete class), it is unknown if a polynomial-time solver is attainable. This complexity gap, known as the P vs. NP problem (Cook, 2003), remains one of the biggest open computer science questions to date (2023).

Recognizing the challenges/unattainability of polynomial complexity solvers, many researchers are focusing on reducing the exponential complexity of known solvers using heuristic functions (Pearl, 1984). One prominent example of an optimization algorithm that utilizes such heuristics is the A* algorithm (Hart et al., 1968). A line of publications (Goldenberg et al., 2014; Felner et al., 2018) exhibited this algorithm's ability to achieve exponential reductions in computational time when paired with an informative heuristic function. Moreover, it is easy to show that given a completely informed heuristic function, A* can solve NP-H problems in time complexity that is polynomial in the solution cost (depth) and the branching factor. This fact has two major implications that are discussed in this paper.

1. Attaining a sufficiently accurate heuristic function estimator (fitted to a completely informed heuristic function) could enable scalable solutions to a large class of NP-H problems.

2. If NP $\not\subseteq$ P/poly then the memory complexity for such sufficiently accurate heuristic function estimator grows super-polynomially with instance sizes.[1]

We show that obtaining a sufficiently accurate heuristic function estimator that represents a completely informed heuristic function is possible using common machine learning approaches (specifically, universal function approximators (Hornik et al., 1989)) for problems that can be reduced to a discrete search space. This fact along with implication 1 motivates a line of publications (McAleer et al., 2018; Agostinelli et al., 2019; 2021b) to train such approximators to estimate completely informed heuristic functions. However, we assert that, given implication 2, these methods are not 'scalable' in nature. That is, the memory size of boolean circuits (Arora & Barak, 2009) required to represent sufficiently accurate heuristic estimators scales super-polynomially with increasing instance sizes. Our formal discussion places previous publications in an appropriate context. Namely by showing that the applicability of heuristic approximations is inherently limited due to scalability barriers.

Complementing our theoretical claims, we provide experiments examining the minimal fully connected neural network that is required to fit a completely informed heuristic function of three NP-H problems with state-of-the-art gradient-based optimization methods to various levels of precision. These results expand on the initial discussion on the scalability of heuristic estimators (Pendurkar et al., 2022a;b) by providing theoretical justifications. The reported results suggest that, as expected, the minimum number of required parameters grows quickly with the instance size.[2]

## 2 Preliminaries

The class of problems that are solvable in polynomial time by a deterministic Turing machine is denoted by P. NP is the class of problems for which a solution can be verified in polynomial time. These problems are solvable in polynomial time by a non-deterministic Turing machine. NP-Hard (NP-H) is the class of problems to which every problem in NP can be reduced in polynomial time. Further, NP-Complete (NP-C) is a class of problems that belong to both NP and NP-H. P/poly is the class of problems that can be solved by polynomial-sized (boolean) circuits. Similar to the P = NP question, NP $\subseteq$ P/poly is another important open question; the Karp-Lipton Theorem (Karp & Lipton, 1980) established that NP $\subseteq$ P/poly would imply the collapse of the polynomial-time hierarchy. Following this result, we assume that NP $\not\subseteq$ P/poly as it is widely believed that polynomial-time hierarchy does not collapse (Arora & Barak, 2009).

**Assumption 1.** *NP $\not\subseteq$ P/poly*

A large set of NP-H problems can be reduced to a *least cost path* problem over an appropriate graph, $G = \{V, E\}$.[3] In such cases, each edge ($e \in E$) is affiliated with a positive cost ($c : E \mapsto \mathbb{R}^+$). Consequently,

---

[1]This also has implications on the time complexity, as in general, time complexity is no less than the memory complexity.

[2]For our code and dataset see `https://github.com/Pi-Star-Lab/unscalable-heuristic-approximator`.

[3]The reduction function is specific to each domain (Bulteau et al., 2015; Cormen et al., 2009; Gupta & Nau, 1992).

the least cost path is defined as $p = \arg\min_{E' \subseteq E} \sum_{e \in E'} c(e)$ subject to $E'$ is an ordered set of edges leading from a given start vertex ($v_s \in V$) to a defined goal vertex ($v_g \in V$). The resulting $p$ is a solution to the original NP-H problem. Note that, multiple optimal solutions to the objective can exist. Despite this, the least cost path problem is not NP-H as, for all known reductions from a NP-H problem, the size of the resulting graph scales exponentially with the problem instance size. That is, the reduction to the least cost path problem is not of polynomial complexity.

The resulting least cost path problems are commonly solved using the A* algorithm. A* searches for the least cost path over the graph without constructing the full graph explicitly. The cost function $g : V \mapsto \mathbb{R}^+$ estimates the cumulative cost from the start vertex to any vertex $v \in V$. A* is also guided by a heuristic function, $h : V \mapsto \mathbb{R}$, defined distinctly for each problem, that estimates the least cost path from any vertex in the search graph, $v \in V$, to a goal vertex. We denote the cost of any valid path $p'$ from a given start vertex to the goal vertex as $cost(p')$. Note, the *cost* function is considered a part of a least cost path problem. The vertices of the search graph are denoted *states* hereafter, and the set of all states is denoted by $S$.

For each state it *generates*, the A* algorithm stores the least cost of reaching it from the start state, that is the $g$ value. A* stores a list denoted 'fringe' of generated states that are not yet *expanded*. At each iteration, A* *expands* a state from its fringe with minimal $f(s) = g(s) + h(s)$ value. If the expanded state is not the goal, then each of its yet unvisited neighbors is *generated*, i.e., it is added to the A* fringe along with its computed $g$ and $h$ values. A* was shown (Pearl, 1984) to *expand* the minimal number of vertices that is required for finding and proving an optimal (least cost) solution.[4]

Given a completely informed heuristic function $h^* : V \mapsto \mathbb{R}$, where $h^*(s)$ is the true minimal cost between $s$ and a goal state, A* would expand only the states along the least cost path. This holds when the ties in the $f$-values are broken in favor of higher $g$ values. That is, the complexity of computing the least-cost path is polynomial in the solution cost and *branching factor* (the maximum degree of the vertices in the search graph). As such, attaining $h^*$ is highly beneficial. While current, domain-specific heuristic functions were shown to result in orders of magnitude speedups (Helmert & Mattmüller, 2008) for NP-H problems, they lack sufficient accuracy to allow polynomial time complexity. Attempting to close this gap, a recent body of work suggested fitting a universal function approximator (Higgins, 2021), e.g., deep neural networks, to $h^*$.

## 3 Related Work

### 3.1 Heuristic search and machine learning

One of the initial works on fitting a completely informed heuristic function was done by Arfaee et al. (2010) which we refer to as *Bootstrap Learning Heuristic* (BLH). They proposed an iterative method ('bootstrap learning') that starts with a known baseline (usually very weak) heuristic estimator and performs a search with A* while storing the expanded states along with their minimal cost to the goal ($h^*$). The resulting dataset is used to fit a new heuristic estimator in a process that continues iteratively until convergence. Recently, McAleer et al. (2018) proposed to apply reinforcement learning (RL) algorithms on top of a *Monte Carlo Tree Search* (MCTS) to solve *Rubik's Cube* instances. They learn the value function (negative of the heuristic function) using temporal difference learning (bootstrapping), with a neural network as the value function approximator, and MCTS as the search algorithm. Following, Agostinelli et al. (2019) presented the *DeepCubeA* algorithm which learns a heuristic function similar to (McAleer et al., 2018), but replaces MCTS with weighted A* (Ebendt & Drechsler, 2009) and uses an adjusted state distribution to perform *bellman updates*. To generate the training samples, they propose taking random walks from the goal state, and updating all the states visited along the path with a bootstrapping method. DeepCubeA presents state-of-the-art results on various NP-H domains like *Rubik's cube*, *Sliding Tile Puzzle*, *Sokoban*, and *Lights Out* in terms of solution quality and run time complexity (number of generated states).

Another line of work focused on learning a policy (Orseau et al., 2018; Orseau & Lelis, 2021), where a policy is a function that maps states (vertices in the search graph) to operators (edges in the search graph). The optimal policy maps states to edges that follows the least cost path to the goal state. Orseau et al. (2018)

---

[4]This claim assumes an *admissible* and *consistent* heuristic function (Pearl, 1984).

proposed using a policy-guided search algorithm (denoted LevinTS) and provided theoretical guarantees on the number of states expanded. *Policy-guided Heuristic Search* (PHS) (Orseau & Lelis, 2021) uses a heuristic function along with a policy. PHS presented a strictly better upper bound on expansions compared to LevinTS under certain conditions, as well as better empirical results for more general cases.

### 3.2 Universal function approximators

There are various universal approximators that can approximate any well behaved function to arbitrary precision. A 2-layer feedforward neural network, with a non-polynomial activation function and a sufficient number of neurons in the intermediate hidden layer, is one such universal function approximator (Cybenko, 1989; Hornik et al., 1989; Pinkus, 1999). Another universal function approximator is a neural network with a non-affine continuous activation function and a fixed number of neurons per layer but with a sufficient number of such layers (Kidger & Lyons, 2020). In the experimental section, we consider these two representative neural network structures as universal approximators.

### 3.3 Complexity of STRIPS planning

Bylander (1994), initially proved that determining the existence of a solution for a given least cost problem with STRIPS formulation is PSPACE-complete. Further, Jonsson (1999) showed that the least cost path problem (formulated as propositional planning instance) is PSPACE-hard with tight approximability bounds. Aghighi et al. (2016) showed approximating the delete relaxation heuristic (within approximability bounds) requires an exponential amount of time under the assumption that the exponential time hypothesis is true. We extend these previous results for problem-specific sufficiently accurate heuristic estimators that are fitted to completely informed heuristic functions.

## 4 The Feasibility of Estimating $h^*$

Given the universal function approximation theorem (Hornik et al., 1989), fitting a universal function approximator to $h^*$ seems promising. That is, given $h^*$ with discrete input (state) space and a bounded function range, a universal function approximator (like a two-layer neural network) can fit $h^*$ to arbitrary precision. A* algorithm with such a fitted universal function approximator as a heuristic function would run in time complexity that is polynomial in the optimal solution cost and branching factor. This is because, when breaking ties in favor of larger $g$ values, A* would expand only the states along an optimal path while generating their successors. Despite this positive result, Theorem 1 shows that memory required for any sufficiently accurate heuristic estimator (represented by boolean circuits) grows super-polynomially. To prove Theorem 1, we first formally define the promise (decision) problems corresponding to (NP-H) least cost path problems.

**Definition 1** (k-cost-path). *A least cost path problem is a promise problem given by a (possibly strict) subset of instances of the decision problem k-cost-path$(C, s, g, k)$, where $s \in S \subseteq \{0,1\}^n$ for some n, a goal state, $g \in S$, and k is an integer represented in unary ($1^k 0$), where*

1. *C is a Boolean circuit computing a function $\{0,1\}^n \to \{0,1\}^{dn}$ representing the expansion operation by mapping a state to the list of at most d adjacent states.*

2. *k-cost-path$(C, s, g, k)$ is a YES instance if there is a valid path p from s to g where $cost(p) \leq k$*

3. *there exists at least one valid path from s to g.*

*The NP-H least cost path problems are least cost path problems that contain the image of a polynomial-time mapping (Karp) reduction for each problem in NP.*

As examples, please see the decision versions of the cost minimization problems pancake sorting, traveling salesman, and blocks world, which are NP-C (See Bulteau et al. (2015), Cormen et al. (2009), Gupta & Nau (1992) respectively). Note that we may view costs that are multiples of some known unit $\epsilon > 0$ as integers without loss of generality.

**Definition 2** (sufficiently accurate heuristic function estimator)**.** *A heuristic function estimator $\hat{h}$ is defined to be sufficiently accurate for a least cost path problem of a given instance size if the following condition holds:*

$$\max_{s \in S} |h^*(s) - \hat{h}(s)| < 1/2$$

*where all edge costs are assumed to be integers.*

A sufficiently accurate heuristic function estimator is equivalent to a completely informed heuristic function and can be obtained by rounding $\hat{h}$ to the nearest integer.

**Lemma 1.** *If a problem $F \in$ NP-C, also belongs to P/poly then NP $\subseteq$ P/poly.*

*Proof.* If $F$ is in NP-C, then by definition for any problem $B \in$ NP, there exists a Karp reduction from $F$ to $B$. Thus, if $F \in$ P/poly then NP $\subseteq$ P/poly. □

**Theorem 1.** *Given Assumption 1, the size of the circuits required for a sufficiently accurate heuristic function estimator for every single NP-C k-cost-path problem grows super-polynomially with the instance sizes.*

*Proof.* By contradiction. Assume a sufficiently accurate heuristic function estimator can be represented by polynomial size circuits. Given this fact, the A* algorithm with a sufficiently accurate heuristic function estimator can find a solution in polynomial time. Thus, the problem is in P/poly. However, given a NP-C k-cost-path also belongs in P/poly, Lemma 1 implies that NP $\subseteq$ P/poly. This fact contradicts our Assumption 1 of NP $\nsubseteq$ P/poly. Thus, the size of a sufficiently accurate heuristic function estimator must grow super polynomially with the instance sizes. □

Theorem 1 implies that a sufficiently accurate heuristic estimator is not scalable in nature. For example, consider an artificial neural network as a universal function approximator. Querying such a network using forward propagation (a single evaluation of $h(s)$ for any $s \in S$) has a complexity of $O(n^2 l)$ where $n > 1$ is the maximal layer width and $l \geq 1$ is the number of hidden layers. Thus, along with the size of the neural network, the querying complexity grows super-polynomially with instance sizes.

### 4.1 Is such high precision truly required?

Heuristic estimation results from previous work (Agostinelli et al., 2019) suggest that these algorithms might not converge to the completely informed heuristic function. As depicted in Figure 3 (on the right-hand side) within the study conducted by Agostinelli et al. (2019), there is evident evidence of an underestimation bias present in the learned heuristic estimator. Nonetheless, the DeepCubeA algorithm reports solving the Rubik's cube while expanding fewer states compared to state-of-the-art, domain-specific heuristics (denoted as PDBs[+] in that paper). These results lead to the following question. "Could it be that one does not need to attain a heuristic estimator with high precision to get sufficiently good (polynomial) A* runtime? And, if so, then perhaps these heuristics are easier to learn?"

The answer to the first question is 'No' for the general case. Helmert & Röger (2008) specifically addresses this issue. They show that "general search algorithms such as A* must necessarily explore an exponential number of search nodes even under the optimistic assumption of almost perfect heuristic estimators, whose heuristic error is bounded by a small additive constant". This implies heuristic estimators that are not accurate enough result in exponential time complexity with search algorithms like A* regardless of the ease of learning to such precision.

### 4.2 Loss function

The training process of a function estimator is usually defined as an optimization problem over an appropriate objective/loss function. Given that all edge costs are multiple of constant $\epsilon > 0$, a parameterized ($\theta$) heuristic

estimator that is sufficiently accurate (Definition 2), over a set of finite states, $D = \{s_1, ..., s_n\}$ is achieved when Eq 1 equals zero.

$$l(D;\theta) = \frac{1}{|D|} \sum_{s \in D} \begin{cases} 0 & \text{if } |\hat{h}_\theta(s) - h^*(s)| < \epsilon/2 \\ 1 & \text{otherwise} \end{cases} \tag{1}$$

Notice that Eq 1 returns the fraction of states with a heuristic approximation error larger than the allowed ($\epsilon/2$) bound. We are not aware of a closed form solution that solves $\arg\min_\theta l(D;\theta)$. Moreover, given that $l$ is not differentiable (around $|\hat{h}_\theta(s) - h^*(s)| = \epsilon/2$) and has no meaningful gradients ($\nabla l = 0$ for its entire domain), it is impractical to apply gradient descent (GD) optimization methods, which are common for training neural networks. As a result, we propose a novel $\epsilon$-*bounded error loss function* denoted $l_\epsilon$. We construct this loss function to be (a) differentiable, (b) provide meaningful gradients, and (c) have optimal solutions (minimums) that align with the true loss function, $l$ (Eq 1). Our proposed $l_\epsilon$ is described in Eq 2.

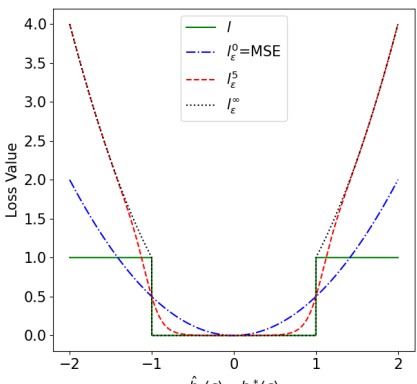

Figure 1: Loss values ($y$-axis) for different loss functions with respect to errors in the heuristic approximation ($x$-axis). The green line is the true loss function, $l$. $l_\epsilon^c$ shows loss values for $l_\epsilon$ loss with specific value of the $c$ parameter.

$$l_\epsilon(D;\theta) = \frac{1}{|D|} \sum_{s \in D} x^2 \left( \frac{1}{1 + \exp\left(-c\left(\left(\frac{2x}{\epsilon}\right)^2 - 1\right)\right)} \right) \tag{2}$$

$l_\epsilon$ is the product of squared error, denoted by $x^2 = (\hat{h}_\theta(s) - h^*(s))^2$ and a sigmoid activation function that tends to 1 as the absolute error exceeds the error bound ($(2x/\epsilon)^2 > 1$) and tends to 0 otherwise. $c$ is a hyperparameter that can be used to change the smoothness of the loss $l$ around the error bound value $\epsilon/2$.

Figure 1 illustrates three loss variants as a function of the estimation error ($x = |\hat{h}(s) - h^*(s)|$) with $\epsilon/2 = 1$. It is easy to see that both loss functions $l$ (Eq 1) and $l_\epsilon$ (Eq 2) are optimized (minimized) when $l_\epsilon(D;\theta) = l(D;\theta) = 0$. These plots also demonstrate the tradeoff presented by the $c$ hyperparameter. When setting $c = \infty$, any solution ($\theta$) that results in $l(D;\theta) = 0$ must also result in $l_\epsilon(D;\theta) = 0$ and vice versa. However, such a $c$ value will produce extreme ($+\infty$, $-\infty$, or 0) gradient values which commonly lead to inefficient or even diverging learning. On the other hand, setting $c = 0$ will lead to the classical MSE loss which is strictly convex and usually results in more stable learning yet has a single optimum (minimum) hence it might fail to detect solutions for the original loss function ($l = 0$).

## 5 Experiments

The empirical study aims to investigate the minimal size of the deep neural networks (as universal function approximators) required to fit a fixed sized dataset of $(s, h^*(s))$ pairs with gradient-descent-based optimization methods. Our evaluation focuses on neural networks as universal function approximators following recent publications that leverage neural networks as the heuristic function (Agostinelli et al., 2019; 2021a; Orseau & Lelis, 2021).

Let $n$ denote the size (number of parameters) of a neural network used to estimate an informed heuristic function. The experimental study is specifically designed to address the following questions.

1. How does $n$ scale with the instance size of NP-H problems?

   (a) Does such an estimator exhibit ineffective learning due to memorization of the training data (overfitting)?

2. Does the scalability trends of $n$ change with variation in the target approximation precision?

3. Does the scalability trends of $n$ change with the choice of loss function?

## 5.1 Experimental limitations

Providing empirical evidence supporting the theoretical claims is challenging due to practical limitations. The limitations are listed below:

- Obtaining the minimal size of a sufficiently accurate neural network is challenging. This is because computing the global optimal parameter assignment for a non-convex optimization problem within practical computation limitations is still an open question.

- Training on the entire state-space of NP-H search problems is impractical because (a) the size of the state space grows exponentially and (b) attaining labels ($h^*$ values) for the entire state space is impractical for NP-H search problems as the time required to optimally solve them (using known algorithms) grows exponentially with instance size.

Given these limitations, we report an evaluation of the minimal size of a sufficiently accurate neural network, which is (1) based on common (gradient-based) optimization techniques; (2) trained on fixed size datasets as opposed to training on the entire state space. Given this, the experimental section should not be viewed as empirical support for our theoretical claims but rather as a study of the impact of our theoretical claims on neural network based heuristic estimators trained over fixed sized datasets as commonly used in literature (Agostinelli et al., 2019; Orseau et al., 2018; Orseau & Lelis, 2021).

## 5.2 Universal function approximators

As universal function approximators, we use the two artificial neural network structures described in Section 3.2. First, a 2-layer neural network (1 hidden layer), with *Rectified Linear Unit* (ReLU) (Nair & Hinton, 2010) as the activation function.[5] We refer to this approximator as 'fixed depth' where the number of neurons in the hidden layer may vary, but the number of layers is fixed. Telgarsky (2016) showed the benefits of exponential reductions in a neural network size can be obtained when considering deeper networks. As a result, our second setting considers a fixed width per layer but allows any number of layers. As in the previous setting, we use ReLU as the activation function for each layer. Similarly, we refer to this structure as 'fixed width'. We used residual connections (He et al., 2016)[6] and batch normalization (Ioffe & Szegedy, 2015) to mitigate issues such as vanishing gradients that we observed during optimization. Our fixed width architecture follows that of Agostinelli et al. (2019), with two differences. First, we reduce the number of neurons per layer to allow a more gradual increase in the number of parameters following the addition of layers. Following theoretical constraints on the minimal layer size presented by Kidger & Lyons (2020), the size of each layer was set as the number of input dimensions + 3. Second, we have one or two linear layers before the residual blocks (as opposed to a fixed number of two layers). This enables us to have an odd as well as even number of layers. Doing so also contributes to a more gradual increase in the number of parameters.

Following previous work (Arfaee et al., 2010; Agostinelli et al., 2019; 2021b) we only consider the 'fixed depth' and 'fixed width' structures. Other variants of neural networks, like convolutional neural networks used by Orseau & Lelis (2021), are not considered as they make specific assumptions regarding the state feature space (e.g., spatial locality) which do not hold in the domains used in our experiments.

## 5.3 Setup

**Fitting Criterion:** For a model to be considered as 'fitting', we train the model on a fixed size training dataset until a certain number of epochs and check whether it follows a 'fitting criterion'. For the first set

---

[5]ReLU activation function satisfies the properties required by the universal function approximation theorem (Sonoda & Murata, 2017).

[6]Residual networks are also universal function approximators (Lin & Jegelka, 2018).

of experiments, we use $l = 0$ (Eq. 1) as the fitting criterion and set a maximum of 3,000 epochs to achieve it. For the second set of experiments, we use $l_\epsilon \leq T$ with $c = 0$ (i.e., MSE) as the fitting criterion and set a maximum of 300 epochs to achieve it (faster convergence was observed for the relaxed criterion when compared to the $l = 0$ criterion). We use different threshold values ($T$) to relax the fitting criterion. This relaxation is required because perfectly fitting the data ($l = 0$) usually requires extreme neural network sizes which grow quickly with the size of the training sets. Note that, for the second set of experiments, the fitted models are not guaranteed to be admissible.[7]

**Loss Function:** For the initial experiments we use $l_\epsilon$ (Section 4.2) as the loss function. We later, use $l_\epsilon$ with $c = 0$ (i.e., MSE) as the loss function as we relax the fitting criterion. MSE loss results are of interest as this loss function was commonly used in previous work (Agostinelli et al., 2019; 2021a;b).

**Datasets:** For all domains, the training set ($D$) included 3,000 random training samples, each of the type $(s, h^*(s))$, in cases where the fitting criterion was set to $l = 0$. When the fitting criterion was relaxed, the training set was set to $8 \times 10^5$ samples. The discrepancy in training set sizes is due to scalability issues when considering fitting to the strict fitting criterion ($l = 0$). We also sample a separate test set that is used to verify if such estimators suffer from memorization (Question 1 (a)) and thus, minimizing test loss should be not viewed as a goal of the experimental section. For all cases, the test set was set to $2 \times 10^5$ random samples.

**Optimization:** We optimize the neural networks using the Adam optimizer (Kingma & Ba, 2015). Note that Adam is not guaranteed to converge on the global optimum. However, it is widely used in the literature (Agostinelli et al., 2019; Orseau & Lelis, 2021) as it usually leads to "close to optimal" solutions (Bölcskei et al., 2017). To mitigate the impact of local minima, we train each neural network 5 times and report the lowest loss. The entire dataset could fit in the memory, as a result, batch size was set to the size of the dataset.

**Finding the minimum number of parameters to fit $D$:** Note, if a neural network with a 1-hidden layer of size $n$ can fit a given function, it must be that a neural network with $n + 1$ neurons in the hidden layer can also fit (by simply setting the additional weights as 0). Similarly, for the fixed width case, if a neural network with $n$ layers can fit a function, it can also fit it with $n + 1$ layers, by setting the weight matrix of the added layer as the identity matrix. Following these understandings, we perform a binary search on the number of neurons (fixed depth) or layers (fixed width) to effectively approximate the minimum number of neurons or layers required to fit $D$.

## 5.4 Domains

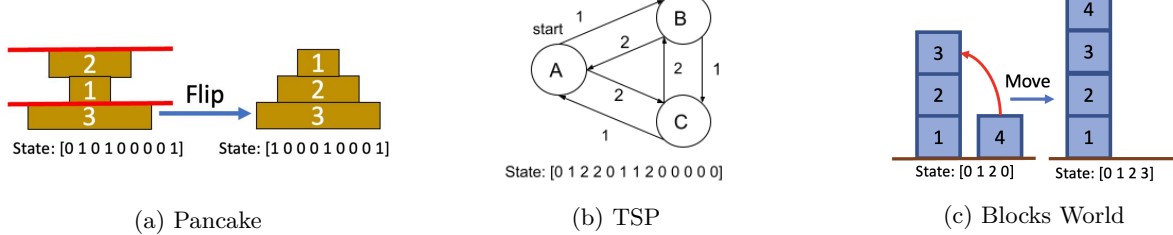

|  |  |  |
|---|---|---|
| (a) Pancake | (b) TSP | (c) Blocks World |

Figure 2: Examples of a problem state for each of three domains along with state encoding $\phi(s)$.

We choose domains in which the number of states grows relatively slowly to have a comparison over a larger range of instance sizes. Following these reasons, we choose pancake sorting (*Pancake*), traveling salesman problem (*TSP*), and blocks world (*Blocks World*). Our choice does not follow previous work which focused on the *Tile puzzle* and *Rubik's cube* domains as the state space for both domains scales with a relatively high exponent. Nonetheless, Tile Puzzle and Rubik's Cube are NP-H (Demaine et al., 2018; Ratner & Warmuth,

---

[7]Admissibility is important to guarantee optimal solutions with search algorithms like A*. A heuristic function $h$ is said to be admissible if $h(s) \leq h^*(s)$ holds $\forall s \in S$.

1990) and their corresponding decision version follows Definition 1, thus the main results from this paper also apply in those domains. We use a domain-specific encoding, $\phi$, and use $(\phi(s), h^*(s))$ to fit the neural network. The cost per operation is multiple of a small constant $\epsilon$ which is 1 for pancake and blocks world, and 0.1 for TSP. An illustrative example of the domains along with their encodings can be seen in Figure 2. Additional details about the domains are in Appendix B.

## 5.5 Results

### 5.5.1 Scaling neural networks for increasing instance size:

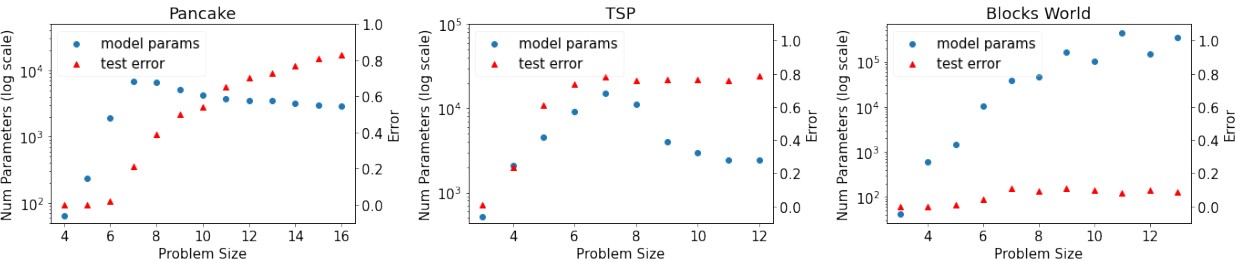

Figure 3: Increase in the minimum number of parameters (log scale) for a 'fixed depth' network architecture required to fit problems with increasing instance sizes on $l_\epsilon$ loss with $c = 10$. On the x-axis, we have the instance sizes for each of the domains. On the y-axis to the left, we have the number of parameters on a log scale. On the y-axis to the right, we have an error which is a fraction of states where $|\hat{h}_\theta(s) - h^*(s)| \geq \epsilon/2$.

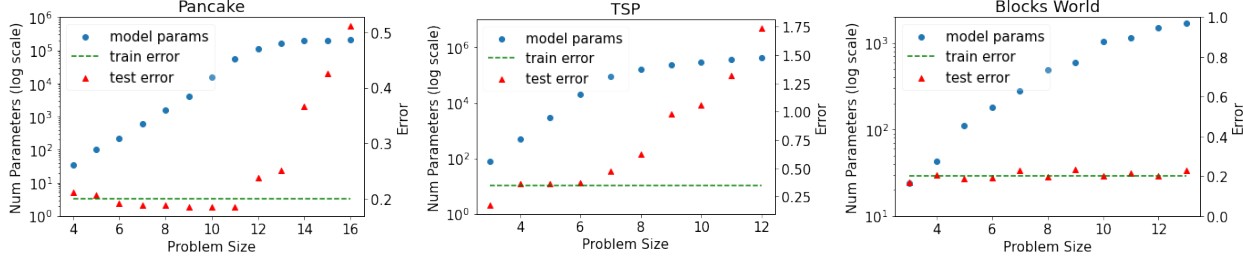

Figure 4: Increase in the minimum number of parameters (log scale) for a 'fixed depth' network required to fit problems with increasing instance sizes on $l_\epsilon$ loss with $c = 0$ (MSE). On the x-axis, we have the instance sizes for each of the domains. On the y-axis to the left, we have the number of parameters on a log scale. On the y-axis to the right, we have MSE values. Loss thresholds are 0.2, 0.35, and 0.2 for pancake, TSP, and blocks world respectively.

To address Experiment Question 1, we report the (empirically) smallest number of parameters (weights and biases) that are able to fit the dataset to $l(D;\theta) = 0$ for increasing instance sizes. Figure 3 shows plots for the number of parameters on a log scale for a fixed depth estimator. The loss function was set to the $l_\epsilon$ loss with $c = 10$. For blocks world, the trend shows that the minimum number of parameters grows quickly as we increase the instance size. The observed trend can be explained by a super-polynomial growth (notice that the y-axis is in log scale), following our theoretical claims. For pancake and TSP, we can see a similar trend until a certain instance size and then a decrease in the minimum number of required parameters. This suggests that, after a certain point, the estimator (neural network) starts overfitting, denoting that the neural network is expressive enough to memorize the entire training set. For instance, if we consider the pancake domain, the rise is steep until instance size 7, and after that it starts decreasing, suggesting memorization of the training samples. We see a similar trend for instance size 7 for TSP.

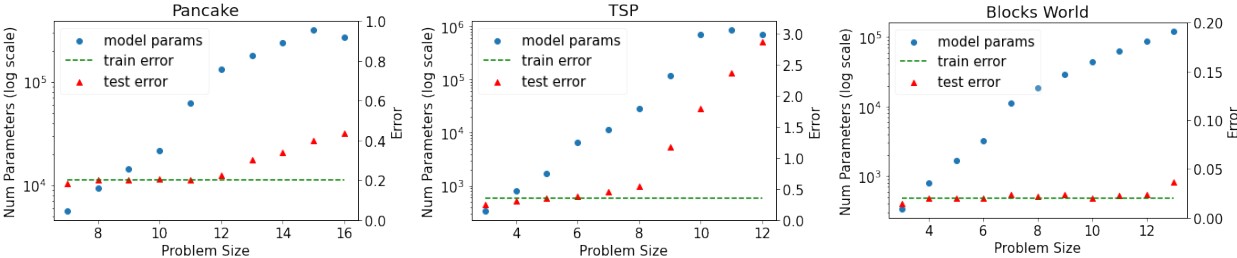

Figure 5: Increase in the minimum number of parameters (log scale) for a 'fixed width' network required to fit problems with increasing instance sizes on $l_\epsilon$ loss with $c = 0$ (MSE). On the x-axis, we have the instance sizes for each of the domains. On the y-axis to the left, we have the number of parameters on a log scale. On the y-axis to the right, we have MSE values. Loss thresholds are 0.2, 0.35, and 0.02 for pancake, TSP, and blocks world respectively.

We observe an unintuitive phenomenon once overfitting begins, the required number of parameters declines with the instance size. We believe that this is due to the added dimensionality in the state space. Our belief follows from (Chen et al., 2007) who state that "any two finite sets of data with empty overlap in original input space will become linearly separable in an infinite dimensional feature space". This statement suggests that with sufficient state dimensions (instance size) even a linear heuristic approximation (neural network with no hidden layers) should suffice for overfitting the training set.

For the rest of the paper, as mentioned in Section 5.3, we relax the fitting condition to $l_\epsilon$ with $c = 0$ (i.e., MSE), and use the same loss for training. Figure 4 shows a plot for MSE with different error thresholds. The trends are very similar to Figure 3, but we see the overfitting starts at later instance sizes for pancake and TSP. We also see an increase in the number of parameters required as we use the larger (1 million) dataset. We see similar patterns for the fixed width case as shown in Figure 5. The plots are noisier than Figure 4, as adding another layer of size $n$ increases the number of parameters by $n^2$ extra weight and $n$ extra bias parameters, making it challenging to observe a continuum of the number of parameters. We can see that the number of parameters required to fit is similar across the fixed depth and fixed width cases ($\sim 5 \times 10^5$) at the point when memorization begins.

To validate our overfitting hypothesis (Question 1a), we include a test error for each estimator size in Figure 3, Figure 4, and Figure 5. In Figure 3 we use Eq. 1 as the test error metric whereas for Figure 4 we use MSE. It is easy to see that while the minimal reported network size stagnates past some instance size (for pancake and TSP), the test error continues to increase which is a common indicator of overfitting. For blocks world, by contrast, we do not see a rise in test error but also no stagnation.

These results also show that neural networks fail to find good latent structures for NP-H problems unlike in text and image based data. This suggests that a sufficiently accurate heuristic function for NP-hard problems might not have a meaningful latent structure and thus require memorization. As a result, the scalability of a heuristic estimator follows that of a naive lookup table, which grows very quickly with the instance size.

### 5.5.2 Invariance to $\epsilon$:

It seems enticing to think that $h^*$ can easily be fitted if the acceptable loss threshold is set high enough. Although it is true that fitting to a larger loss threshold is easier than fitting to a low threshold, we see that, even for larger thresholds, the number of parameters still grow very quickly with the instance size. Figure 6 shows the growth in the number of parameters across instance sizes and for various representative loss thresholds. Although the number of parameters required for lower precision is lower, the required number of parameters still grows very quickly. We can also see that, as we increase the threshold, the point where overfitting begins changes. For instance, for the pancake problem, we see that for threshold 0.6, the curve is growing quickly until instance size 17. Similar to our results in the previous subsection, we begin to see overfitting when we have a minimum of $\sim 5 \times 10^5$ parameters, across different thresholds and domains.

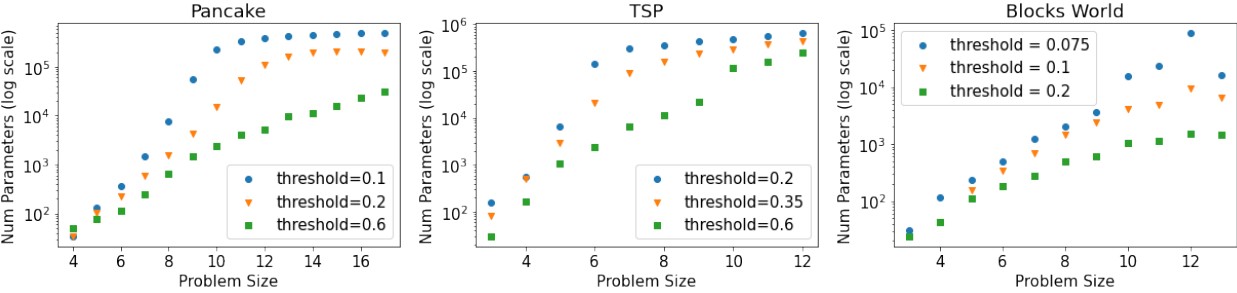

Figure 6: Increase in the minimum number of parameters (log scale) required to fit problems with increasing instance sizes for different loss thresholds on $l_\epsilon$ loss with $c = 0$ (MSE). On the x-axis, we have the instance sizes for each of the domains. On the y-axis, we have the number of parameters on a log scale. For each of the three domains, we use three different thresholds.

These results suggest that the answer to Question 2: "Is the scalability trend of $n$ sensitive to variation in the target approximation precision?" is 'No'.

### 5.5.3 Invariance of unscalability trends to loss objectives:

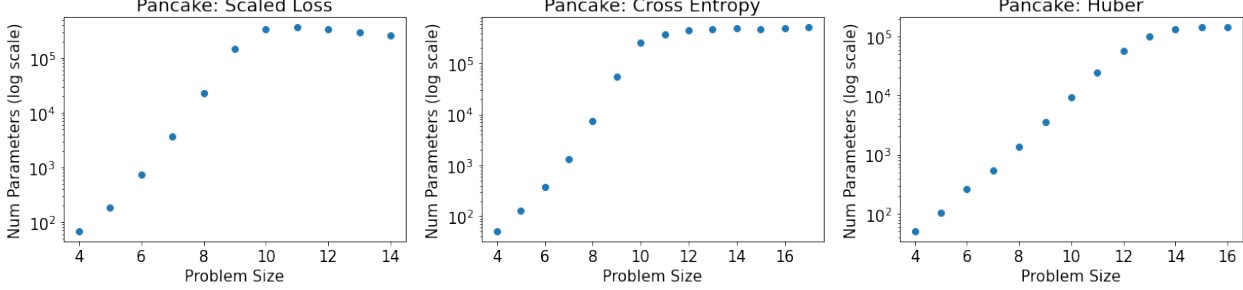

Figure 7: Increase in the minimum number of parameters (log scale) required to fit problems with increasing instance sizes on various loss functions for the Pancake problem. On the x-axis, we have the instance sizes. On the y-axis, we have the number of parameters on a log scale. The thresholds used for scaled loss, accuracy threshold for cross-entropy loss, and Huber loss are 0.001, 90%, and 0.1.

Beyond the initial variants of $l_\epsilon$ loss, we now report results for three additional loss functions.

1. *categorical cross entropy loss* by viewing the heuristic estimation as a classification problem.

2. *Scaled loss* defined as: $\frac{1}{N} \sum_{i=1}^{N} (1 - \frac{\hat{h}_\theta(s_i)}{h^*(s_i)})^2$

3. *Huber loss* (Huber, 1992) with delta $= 1$.

The motivation behind the scaled loss variant is as follows. In many cases, the suboptimality factor of the A* algorithm is governed by the relative error in $\hat{h}_\theta$ and not the absolute error. The *weighted*-A* algorithm Ebendt & Drechsler (2009) is a prominent example.

Figure 7 shows a plot for three loss functions for the pancake problem using the 'fixed depth' network structure (similar trends were observed for the other domains and network structure). These results suggest that the trend of a quick increase in the minimum number of parameters of a fitting neural network followed by stagnation (suggesting overfitting) is agnostic to the choice of the loss functions. Thus, the answer to Question 3 "Is the scalability trend of $n$ sensitive to the choice of loss function?" is 'No'.

## 6 Conclusion

In this paper, we investigate the unscalability of heuristic estimators for NP-hard search problems. We provide theoretical justifications for our claim of the unscalability of universal function approximators as heuristic estimators by showing the size of such sufficiently accurate heuristic estimators must grow super-polynomially if $NP \not\subseteq P/poly$. We support these theoretical results empirically using a common universal function approximator – neural network in two settings. The experiments (with conventional practices) suggest that irrespective of the architecture of a neural network, choice of optimization objective, and required precision, the number of parameters needed to fit a completely informed heuristic function grows quickly with the problem instance size, highlighting the limitations of the current state-of-the-art approaches. The main conclusion drawn from this paper is that heuristic search algorithms that propose to fit the completely informed heuristic function for NP-Hard search problems are inherently not scalable. We expect that our paper will impact the research community by (1) shifting its research efforts to other/additional ways of integrating heuristic search with machine learning (2) introducing scalability as an important criterion of evaluation for future research in the area of heuristic search and machine learning. We believe our study would assist researchers in developing machine learning based approaches that outperform the state-of-the-art classical planning based solutions (for example see Flatland Challenge (Li et al., 2021) where current classical methods are state-of-the-art).

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
