# OpenReview forum: "The (Un)Scalability of Informed Heuristic Function Estimation in NP-Hard Search Problems"
_TMLR — Accepted by TMLR_

### Review · Reviewer_scU8 · 2023-09-07

**Summary Of Contributions:**

In prior theoretical work, $A*$ search was shown to solve NP-hard problems in polynomial time assuming it has access to a completely-informed heuristic function. This result supports the exploration of training neural networks (as universal function approximators) to accurately match a completely-informed heuristic function on a finite training set for a specific class of problem instances. This work examines whether this approach scales to larger and larger problem instances. Theoretically, they show that if this approach does scale, it contradicts expected results in complexity theory (NP $\not\subseteq$ P / poly). Empirically, they show that exponentially larger neural networks are required to learn completely-informed heuristic functions on larger problem instances from a variety of least-cost path problems. They find that this empirical result holds across the choice of loss function used to train the network as well as the desired target precision. They conclude that research should look towards other techniques for scaling to large problem instances.

**Audience:**

Yes

**Broader Impact Concerns:**

I have no broader impact concerns.

**Claims And Evidence:**

Yes

**Requested Changes:**

- I would suggest trying to vary the training set size and number of training epochs.
- I would suggest exploring one additional classical loss, e.g., Huber loss, $\epsilon$-insensitive loss or other.

Minor:
- Can you reconcile section 4.1 against definition 2 and its implication? In other words, why do we not set $\epsilon=1$ always?
- What minibatch size do you use for training the neural networks?
- For your fitting criterion, do you measure loss using minibatches or the full dataset?
- In Figure 5, is the right y-axis measuring MSE values?
- Please add a mention of the pancake problem in Figure 7's caption.

**Strengths And Weaknesses:**

Strengths:
- The writing is clear and the theoretical statements tell a story that is paired well with empirical results.
- The proposed loss function is sensible, intuitive, and novel to my knowledge.
- The experiments specifically address the questions raised by the authors.

Weaknesses:
- The neural network training does not appear to vary two important dimensions that can improve the learned model's performance: 1) training set size and 2) number of training epochs. It would make intuitive sense that larger problem instances need larger networks, larger training sets, and longer training times to succeed.
- There are other classical loss functions from the literature that may make sense here. I do not expect them to change the qualitative results, but they are an alternative to inventing your own: a) Huber loss and b) $\epsilon$-insensitive loss (used for support vector regression).

The authors conclude that this work suggests exploring an alternative approach, but it is unclear if their findings are enough to suggest they would generalize to other problem classes. In some sense, substituting neural networks for completely-informed heuristic functions pushes the problem of heuristic function design to the neural network level of architecture design and training protocols. Neural network training is NP-hard itself (i.e., it is generally a non-convex optimization problem), and so we are trading one NP-hard problem for another. That being said, if one takes the time and has the expertise to carefully design and train a neural network for a given task, the past decade has shown numerous empirical success (e.g., the deep-RL references you cite) on enormous problem instances. Why argue against an approach that has already seen great success on large problem instances?

---

> ### Author Response · Authors · 2023-09-12
> **Response to Reviewer scU8 - Major Comments**
>
> I would suggest trying to vary the training set size and number of training epochs.
> - Answer: (1) training set size: We conducted experiments that involved varying the size of our training dataset, which correlated with increasing instance sizes, as recommended. Specifically, we attempted to assemble a dataset that contains approximately x% of the entire state space. However, we encountered a significant challenge in this endeavor. The number of states grows exponentially with the size of the instances, leading to an exponential growth in the size of the dataset. Additionally, acquiring labels for instances with known optimal solvers becomes increasingly arduous as the instance sizes grow. Due to these limitations, we regrettably were unable to compile datasets for the Traveling Salesman Problem (TSP) and the Blocks World, preventing us from conducting a comprehensive study of scalability trends within a practical timeframe and memory constraints. However, it's worth noting that we achieved relatively effective results for the Pancake Problem as we leverage the gap heuristic [a]. To address the reviewer's concern, we have included these results in Appendix C. As anticipated, we observed that the number of parameters required exhibits a super-polynomial growth rate.
> - Answer: (2) number of training epochs: We tuned the maximum number of iterations to 3,000 for the initial set of experiments and 300 for the later set of experiments. This number was chosen as it ensured that the Adam optimizer would successfully converge, even when dealing with the most challenging datasets, all within the provided computational limits.
> It's important to clarify that our primary research focus does not revolve around the convergence speed relative to instance size. Instead, the core of our study is dedicated to substantiating the existence of a neural network weight assignment that can effectively represent a sufficiently accurate heuristic function estimator. Consequently, we believe that delving into an extensive study of convergence speed, per instance size, would not align well with our primary research objectives and may divert readers' attention from the key contributions of our work.
>
> I would suggest exploring one additional classical loss, e.g., Huber loss,  \epsilon-insensitive loss or other.
> - Answer: As per suggestion, we have replaced the last subfigure (MSE loss) in Figure 7 with Huber loss with delta=1. The general trends, previously reported, persist. Following this, we have updated the text in Section 5.4.3 accordingly.
>
>  if one takes the time and has the expertise to carefully design and train a neural network for a given task, the past decade has shown numerous empirical success (e.g., the deep-RL references you cite) on enormous problem instances. Why argue against an approach that has already seen great success on large problem instances?
> - Answer: While deep reinforcement learning and deep learning have excelled in various domains, we must note that their success does not consistently extend to solving NP-hard search problems, the subject of our paper. For instance, classical planning methods outperformed RL in the NeurIPS 2020 Flatland Challenge [b] by significant margins. Similarly, DeepCubeA exhibits scalability issues, underestimating heuristic values as random walk lengths increase (Figure 3 in the original paper). Our paper's main goal is to highlight these inherent challenges, both theoretically and empirically. We hope this prompts the ML and search communities to focus on scalable solutions for NP-hard problems to replicate the success seen in other areas. One such direction could be focusing on learning admissible heuristic functions (such that h(s) <= h*(s) for all s in S).
>
>
> [a] Valenzano, Richard, and Danniel Yang. "An analysis and enhancement of the gap heuristic for the pancake puzzle." Proceedings of the International Symposium on Combinatorial Search. Vol. 8. No. 1. 2017.
>
> [b] Li, Jiaoyang, et al. "Scalable rail planning and replanning: Winning the 2020 flatland challenge." Proceedings of the international conference on automated planning and scheduling. Vol. 31. 2021
>
>
> Please refer to our revised version for the images.

---

> > ### Comment · Reviewer_scU8 · 2023-09-28
> > **Major Comments - Follow-Up**
> >
> > Thank you for your efforts running additional experiments and explaining your motivations. I am satisfied with the additional experiments and your answer. I do think your paper would benefit from including a similar discussion citing, for example, the Flatland Challenge you referenced.

---

> > > ### Comment · Reviewer_scU8 · 2023-09-28
> > > **Minor Comments - Follow-Up**
> > >
> > > Yes, sorry, I meant section 4.2 (equation 1) versus definition 2. Definition 2 says a heuristic is sufficiently accurate if the absolute heuristic error is below $1/2$ while equation 1 penalizes a heuristic having error above $\epsilon/2$ with $\epsilon \ll 1\textemdash$ Figure 1 says the loss threshold ($\epsilon$) is 0.2, 0.35, and 0.2 for pancake, TSP, and blocks world respectively. Why set $\epsilon < 1$ if theory says $\epsilon=1$ is sufficient?

---

> ### Author Response · Authors · 2023-09-28
> **Response to Reviwer scU8's follow up**
>
> Major: We have added this discussion to the Conclusion following the suggestion.
>
> Minor: We believe there is confusion here. We believe the reviewer is referring to Figure 4. Note, \epsilon does not equal the loss threshold. \epsilon is a known unit such that the edge costs are multiples of such a \epsilon > 0 (1 for Pancake and BW and 0.1 for TSP). This is stated in Section 4 “Note that we may view costs that are multiples of some known unit epsilon > 0 as integers”. The loss thresholds are the values of the loss function (Equation 2) with \epsilon values set (as described above) and c values (c=10 for Figure 3, c=0 when referring to MSE). Given the confusion, we clarified this in Section 5.3.

---

> > ### Comment · Reviewer_scU8 · 2023-10-02
> > **Minimum Cost Difference vs Loss Thresholds**
> >
> > Yes, sorry for the confusion again, I meant Figure 4.
> >
> > Ah okay, I understand now that you can rescale costs w.l.o.g. such that costs are unitary and then $\epsilon=1$ makes sense; equivalently, you can leave them as is and set $\epsilon$ to be the minimum difference between cost levels.
> >
> > And I see you have defined loss threshold $T$ in Sec 5.2, $\ell_{\epsilon} \le T$.
> >
> > Let me know if I'm still misunderstanding something, otherwise, I'm clear. Thank you.

---

> > > ### Author Response · Authors · 2023-10-04
> > > **Acknowledging Reviewer scU8's comment**
> > >
> > > Your explanation is accurate. Thanks for the feedback.

---

### Review · Reviewer_oyZD · 2023-09-19

**Summary Of Contributions:**

The paper shows that if a NP-hard problem is encoded as a pathfinding problem on a graph of super-polynomial size, then learning a good enough heuristic function to solve the problem optimally in polynomial time requires a deep neural network of super-polynomial size. The authors define a custom-made loss function that closely approximates the real loss function, but is differentiable. Experiments on three NP-hard problems show the increase in size and error.

**Audience:**

Yes

**Claims And Evidence:**

Yes

**Requested Changes:**

Throughout the paper, the style used to cite papers is not readable, because the list of authors is not placed between brackets, but the paper itself is not written to accommodate this style of citation. Please fix this.

Page 1, implication 2, why is there an implication on the memory complexity, and not on the time complexity? Is the assumption that the heuristic function is only looking up a value in a table, or must be a large DNN? Please clarify this point.

At this point of my read (end of page 2), there seems to be an assumption that there is a single least-cost path. However, this is not true in general, and I’m worried it may affect the results. If this is an assumption that can be made without loss of generality, please make it clear from the beginning. (I see on page 4 that breaking ties in favour of larger g values is mentioned, but no explicit link is made).

After reading Section 3, I am left wondering whether DNNs provide an admissible and consistent heuristic. If yes, how? If no, is it a problem? If no, why? If yes, how is it handled? Furthermore, it may be useful to define admissible and consistent.

In Definition 1, the cost function is used in the definition to qualify a YES-instance, but it is written that it is part of the input.

I think it’d be best not to refer to Figure 3 on page 5, as it is on page 8.

**Strengths And Weaknesses:**

The results are not surprising, but they are worth writing down.

The paper is relatively clear, although, due to my background, I do not understand all the discussion about the different DNN design choices. However, it is reasonable for TMLR.

There are a few pages of results. The discussion does not clearly convey why they are all meaningful.

---

> ### Author Response · Authors · 2023-09-28
> **Response to Reviewer oyZD - Part 1**
>
> There are a few pages of results. The discussion does not clearly convey why they are all meaningful.
> - Answer: To clarify the reviewer’s concern we have added the following text at the beginning of Section 5: “Providing empirical evidence supporting the theoretical claims is challenging due to practical limitations. Specifically, (1) we cannot obtain the minimal size of a sufficiently accurate universal function approximator because obtaining global optimal parameter assignment is not feasible for a non-convex optimization problem within practical computation assumptions, and (2) we cannot train on the entire state-space of NP-H search problems because (a) the size of the state space explodes exponentially and (b) attaining labels (h* values) for the entire state space is impractical for NP-H search problems as the time required to optimally solve grows exponentially with instance size. Consequently, the empirical study aims to investigate the minimal size of the deep neural networks (as universal function approximators) to fit a fixed sized dataset of (s, h*(s)) pairs with gradient-descent-based optimization methods. Our evaluation focuses on neural networks as universal function approximators following recent publications that leverage neural networks as the heuristic function (Agostinelli et al., 2019; 2021a; Orseau & Lelis, 2021).”. The experiments suggest that, under conventional practices, the exponential growth of parameters in deep neural networks is necessary to estimate the h* function with deep neural networks as the instance size increases. While it's important to note that these results do not definitively prove our theory, they do show that the limitations we predicted in our theory are indeed visible in the domains we studied even under practical settings.
>
> the style used to cite papers is not readable, because the list of authors is not placed between brackets
> - Answer:  Thanks for pointing out. We have fixed the citation style.
>
> Page 1, implication 2, why is there an implication on the memory complexity, and not on the time complexity? Is the assumption that the heuristic function is only looking up a value in a table, or must be a large DNN? Please clarify this point.
> - Answer: There is indeed an implication with respect to both time and memory. Since the circuit representing the estimator must be large, it also requires a high time complexity to evaluate. Note that an algorithm may have high time complexity without using a large amount of memory, but in general, the amount of memory used is no greater than the time complexity, assuming that only a constant size word can be read or written on each step. Thus the implication that a large representation is required is the more damning. To avoid confusion we have added clarification to the Introduction Section as footnote 1.
>
> At this point of my read (end of page 2), there seems to be an assumption that there is a single least-cost path. However, this is not true in general, and I’m worried it may affect the results. If this is an assumption that can be made without loss of generality, please make it clear from the beginning. (I see on page 4 that breaking ties in favour of larger g values is mentioned, but no explicit link is made).
> - Answer: We would like to clarify that we do not assume that there exists only a single least-cost path. Throughout the paper, we consider the A* algorithm where ties in f values are broken in favor of the g values (also mentioned in footnote 4 page 2 in the original submission). Doing so, allows the A* algorithm to provide an optimal solution in linear time given a completely informed heuristic function even when there exists multiple optimal solutions. We now have updated section 2 Preliminaries to clearly state this.

---

> ### Author Response · Authors · 2023-09-28
> **Response to Reviewer oyZD - Part 2**
>
> After reading Section 3, I am left wondering whether DNNs provide an admissible and consistent heuristic. If yes, how? If no, is it a problem? If no, why? If yes, how is it handled? Furthermore, it may be useful to define admissible and consistent.
> - Answer: In general, DNNs fitted to h* do not provide guarantees that they will return an admissible and consistent heuristic function. Previous papers that attempt to fit neural network to h* also discuss this limitation in their papers. For example, (1) Agostinelli et. al 2019 state that “Although DeepCubeA’s heuristic function is not guaranteed to be admissible, and thus does not satisfy the requirement for these theoretical bounds” (2) Orseau & Lelis 2021 state that “Note that this loss function tends to make the heuristic admissible” (tends suggests no guarantees) (3) Arfaee et al. 2011 state “As in the earlier studies (Ernandes & Gori 2004; Samadi, Felner, & Schaeffer 2008), which may be seen as doing one step of the bootstrap process with a very strong initial heuristic, the learned heuristic might be inadmissible, i.e., it might sometimes overestimate distances”. Note that as these heuristic functions are not admissible, they also lack consistency. Given the results that we present in the paper, we believe one possible further direction would be to develop methods that design loss functions such that the learnt heuristic function guarantees admissibility/consistency under some conditions. Although this research direction is important, the discussion is orthogonal to the claims presented in our paper.
>
> In Definition 1, the cost function is used in the definition to qualify a YES-instance, but it is written that it is part of the input.
> - Answer: Thanks for pointing this out. The cost function is considered to be a parameter defining a least cost path problem; properly, there is a family of least cost path problems, given by various choices of cost function. We do not state it as part of the input explicitly because we are considering problems in which the cost function is fixed – for example, when we cast pancake sorting as a least cost path problem. The qualification that a least cost path problem is NP-complete means that such a problem with a fixed cost function (not part of its input) is NP-complete. We have updated Section 2 Preliminaries paragraph 3 to clarify this further. If we misinterpreted your comment, please let us know; we are happy to discuss this further.
>
> I think it’d be best not to refer to Figure 3 on page 5, as it is on page 8.
> - Answer: We believe there is some confusion here. On page 5 we are referring to Figure 3 (right) in a different paper (Agoestinelli et al. 2019) and not Figure 3 in our paper. We rephrased the statement to avoid confusion.

---

> > ### Comment · Reviewer_oyZD · 2023-10-01
> > **Second round of questions to authors**
> >
> > Thanks for your answers.
> >
> > I have more questions regarding the memory complexity. The paper reads "Theorem 1 shows that memory required for such a sufficiently accurate heuristic estimator grows super-polynomially". However, it's unclear to me what "such a sufficiently accurate heuristic estimator" means here. Specifically, the "such". Because one could imagine a non-NN heuristic estimator that requires super-polynomial time but only polynomial space. So, in my mind, the sentence I quoted should specify that it applies only to a DNN-type estimator, for which inference is polynomial in the size of the network. I think that the result you show could be clearer if it was explicitly decomposed in:
> > 1. Any sufficiently accurate heuristic estimator would require super-polynomial time. (theorem 1 modified to only discuss time requirement of a general heuristic).
> > 2. If a DNN heuristic estimator with inference time that is polynomial in its size is sufficiently accurate, then its size cannot be polynomial in the original instance.
> >
> > On a separate topic, I think it'd be good to recall at the beginning of the experimental section (unless it's there and I missed it) the fact that these heuristics are non-admissible. The theoretical part of the paper discusses "sufficiently accurate heuristic function estimator", which are by definition not admissible, but are just a rounding operation away from being so (as written in the paper). So, essentially, the theoretical part of the paper deals with admissible heuristics. But this breaks when moving to the experimental section. Are we not concerned by the fact that A* will not be optimal, and to what extent that is? Is this visible somewhere in the results?
> >
> > I hope that makes sense. Thanks!

---

> > > ### Author Response · Authors · 2023-10-04
> > > **Authors' response to second of questions by Reviewer oyZD**
> > >
> > > I have more questions regarding the memory complexity. The paper reads "Theorem 1 shows that memory required for such a sufficiently accurate heuristic estimator grows super-polynomially". However, it's unclear to me what "such a sufficiently accurate heuristic estimator" means here. Specifically, the "such". Because one could imagine a non-NN heuristic estimator that requires super-polynomial time but only polynomial space. So, in my mind, the sentence I quoted should specify that it applies only to a DNN-type estimator, for which inference is polynomial in the size of the network. I think that the result you show could be clearer if it was explicitly decomposed in:
> > > Any sufficiently accurate heuristic estimator would require super-polynomial time. (theorem 1 modified to only discuss time requirement of a general heuristic).
> > > If a DNN heuristic estimator with inference time that is polynomial in its size is sufficiently accurate, then its size cannot be polynomial in the original instance.
> > > - Answer: We would like to clarify that when we talk about “memory required for … estimators” we are specifically concerned with memory size of boolean circuits (Barak & Arora 2009) used to represent the estimator (specifically universal function approximators). Note that the evaluation of boolean circuits is known to be P-complete (Barak & Arora 2009) and in particular any function (including a universal function approximator) can be represented by a boolean circuit of size polynomial in its computation time (See Section 9.3 (Sipser) [a]). We have now updated the text to make the claims accurate and avoid any further confusion. Regarding the suggested revisions (points 1 and 2), we are making a more general claim (beyond DNNs).
> > >
> > > I think it'd be good to recall at the beginning of the experimental section (unless it's there and I missed it) the fact that these heuristics are non-admissible. The theoretical part of the paper discusses "sufficiently accurate heuristic function estimator", which are by definition not admissible, but are just a rounding operation away from being so (as written in the paper). So, essentially, the theoretical part of the paper deals with admissible heuristics. But this breaks when moving to the experimental section. Are we not concerned by the fact that A* will not be optimal, and to what extent that is? Is this visible somewhere in the results?
> > > - Answer: As discussed in the original response, the paper is mainly concerned with completely informed heuristic functions (h*) which is, of course, admissible and consistent. For the experimental section, we provide results with fitting criteria of attaining a sufficiently accurate estimator that is admissible and completely informed after rounding (See Figure 3). On top of that, we show that relaxing the fitting criteria to attain (non-admissible) heuristic function with a bounded MSE also results in an exponential growth in the number of parameters of the fitted DNN (Figure 4,5). We see this as strengthening our argument that even under relaxed experimental settings, we observe that a number of parameters of a minimum-sized neural network grow exponentially with instance size. The optimality of the A* algorithm paired with such learnt neural network-based heuristic functions was previously discussed (see Arfaee et al. 2011 Tables 1-11,  Agostinelli et al. 2019 Table 4, 5). Given the conclusions drawn from our paper, we expect the optimality results to get worse (solution cost >> optimal) with increasing instance sizes depending on the bias of the network (that depends on state distribution). In order to address the reviewer’s concern we now clearly state that accurately fitting h* results in an admissible heuristic while the relaxed version is not guaranteed to be admissible.
> > >
> > > [a] Sipser, Michael. "Introduction to the Theory of Computation." ACM Sigact News 27.1 (1996): 27-29.

---

> > > > ### Comment · Reviewer_oyZD · 2023-10-04
> > > > **Tentative final answer**
> > > >
> > > > Thank you for your replies and edits. I have no further questions at the moment.

---

### Review · Reviewer_6bhh · 2023-09-20

**Summary Of Contributions:**

The ability of a heuristic search such as the A* search depends on both the heuristic function and the search algorithm.
If we had a completely informed heuristic function, A* search could solve many NP-hard problems rapidly.
However, Theorem 1, the main result of the first half of this paper, proves that such a heuristic function cannot be computed in polynomial time using polynomial-sized circuits.

In the second half of the paper, the authors evaluated the performance of neural-network-based heuristic functions for several real-world problems. They conducted experiments to evaluate the size of neural networks that achieve a given accuracy as the problem size increases.

**Audience:**

Yes

**Broader Impact Concerns:**

I find no concerns.

**Claims And Evidence:**

Yes

**Requested Changes:**

I want the authors to clarify whether the experimental results and analysis in section 5 are really related to the theoretical results in section 4. However, I do not know what kind of revision would be best.
I have the following question and would appreciate the authors answering it.

In many problems, there is a large bias in the value of $h^*$.
For example, in the 8-puzzle (3x3 sliding puzzle), there are only two states where $h^* = 1$, and the ratio of the states with $h^* <= 10$ is very small.
If we randomly select 3,000 training data from all possible 8-puzzle states, most of them would have an h^* value between 18 and 26 and likely cause overfitting.

However, the bias is not so clear for the 2x2 or 2x3 sliding puzzle.
Thus, the smaller the problem, the less likely it may be to overfit.

In other words, some of the results observed in the experiments in Section 5 may
not be due to the nature of P/poly but are simply a phenomenon caused by the difficulty of training the model due to the bias of the h^* distribution.
To resolve this question, I would like to see the distribution, or at least the histogram, of the h^* of the training data.

**Strengths And Weaknesses:**

Strengths.

- Theorem 1 is an interesting result. The proof is brief but clear and convincing.

- Sections 2 and 3 are a good survey of previous theoretical work on the capabilities of search algorithms.
The explanations are well-organized and easy to understand.

- Probably, this paper without section 5 is already good.

Weaknesses.

- The connection between section 5 and the rest of the paper is not clear.

---

> ### Author Response · Authors · 2023-09-28
> **Response to Reviewer 6bhh**
>
> The connection between section 5 and the rest of the paper is not clear.
> - Answer: Following the reviewer’s concern, we added the following paragraph at the beginning of Section 5, “Providing empirical evidence supporting the theoretical claims is challenging due to practical limitations. Specifically, (1) we cannot obtain the minimal size of a sufficiently accurate universal function approximator because obtaining global optimal parameter assignment is not feasible for a non-convex optimization problem within practical computation assumptions, and (2) we cannot train on the entire state-space of NP-H search problems because (a) the size of the state space explodes exponentially and (b) attaining labels (h* values) for the entire state space is impractical for NP-H search problems as the time required to optimally solve grows exponentially with instance size. Consequently, the empirical study aims to investigate the minimal size of the deep neural networks (as universal function approximators) to fit a fixed sized dataset of (s, h*(s)) pairs with gradient-descent-based optimization methods. Our evaluation focuses on neural networks as universal function approximators following recent publications that leverage neural networks as the heuristic function (Agostinelli et al., 2019; 2021a; Orseau & Lelis, 2021).”. The experiments suggest that with the commonly used practices, the number of parameters of the deep neural network required to fit the h* function grows exponentially with instance size. While cannot be viewed as proof for our theoretical claims, these results demonstrate that the limitations predicted by our theory are indeed visible in the reported domains.
>
> “Bias in the values of the h* values…experiments in Section 5 may not be due to the nature of P/poly but are simply a phenomenon caused by the difficulty of training the model due to the bias of the h^* distribution.”
> - Answer: In order to avoid confusion, we would first like to clarify the notation. Following common machine learning literature we use “bias” to refer to inductive bias (also known as learning bias) i.e., “any basis for choosing one generalization over another, other than strict consistency with the observed training instances.”[2]. To differentiate “bias in h* values” from the inductive bias; we refer to bias in the h* values as label imbalance (similar to the class imbalance problem [1]). Although we agree that there is a label imbalance towards larger h* values, we do not see how this could lead to overfitting. Did you mean underfitting? I.e., always predicting the most common label? While underfitting might be reasonable, our results (test set vs training set accuracy) suggest otherwise (see discussion in Section 5.4.1). Note that following common machine learning assumptions, our train and test set are sampled from the same distribution. If we misinterpreted your comment, please let us know; we are happy to discuss/clarify this point further.
>
> “To resolve this question, I would like to see the distribution, or at least the histogram, of the h^* of the training data.”
> - Answer: We have added these results for Pancake and Blocks World in Appendix D (just for the rebuttal, as we are unsure how to share images on openreview). Based on the histograms, will label imbalance is apparent, we do not observe that the label imbalance gets worse with instance sizes for Pancake and Blocks World. Note that as we consider a weighted graph for TSP where the edge weights are sampled from a uniform distribution, the distribution over h* values is not well behaved for TSP.
>
> [1] Fernández, A., García, S., Galar, M., Prati, R. C., Krawczyk, B., & Herrera, F. (2018). Learning from imbalanced data sets (Vol. 10, pp. 978-3). Cham: Springer.
>
> [2] Mitchell, Tom M. "The need for biases in learning generalizations." (1980).

---

> > ### Comment · Reviewer_6bhh · 2023-10-01
> > **Response to the response**
> >
> > Thank you for the histograms. The experimental results on paper for pancake start with problem size 4, but the histogram of $h^*$ starts from 7. Could the authors show the histograms for 4-6?  Also, does the X-axis cover all possible $h^*$ values? For example, is there no data for h^* lower than 6 for Pancake-15?
> >
> > > "we do not observe that the label imbalance gets worse with instance sizes for Pancake and Blocks World."
> >
> > What I guess is that if we sample only 3,000 training data samples from the original data, the imbalance of the training data could become "better" for larger instances of Pancake and BW.
> > For example, if we sample 3,000 data from BW-5 or Pancake-7, the data would likely include some of the outliers.
> > However, if we sample from BW-13 or Pancake-16, the probability of selecting the outliers becomes smaller.
> >
> > My hypothesis is that for smaller size problems, the 3,000 data samples are more likely to have samples with very small $h^*$ values.
> > And if a model overfits the 3,000 data samples, the more inaccurate it becomes (in terms of overall accuracy).
> > Do the authors think this hypothesis is reasonable?
> >
> > The results in Figure 3-7 are highly affected by these details of the experimental setup. Therefore, I have a concern that these results do not relate to the theoretical analysis provided in section 4.
> > Could the authors elaborate more on this? Frankly, I feel that the results in section 5 are an example of the difficulty of empirically supporting the theoretical claims.

---

> ### Author Response · Authors · 2023-10-04
> **Author's reply to "Response to the response" by Reviewer 6bhh**
>
> The experimental results on paper for pancake start with problem size 4, but the histogram of ℎ∗ starts from 7. Could the authors show the histograms for 4-6?
> - Answer: Apologies for not providing them initially. We intended to focus on the trends visible in the larger instances. We have now added the full results as requested. Please see the updated appendix with the plots.
>
> Also, does the X-axis cover all possible ℎ∗ values? For example, is there no data for h^* lower than 6 for Pancake-15?
> - Answer: We empirically observe that a fixed number of random walks from the goal does not cover all possible h* values (especially for the lesser h^* values). This is because of the h* label imbalance. The sampled set (of size 10^6) had no h* values < 6 for 15-pancake. This is a reasonable outcome as the probability of sampling these states is extremely low (< 14^5/15! \approx 10^-7).
>
> "we do not observe that the label imbalance gets worse with instance sizes for Pancake and Blocks World."
> What I guess is that if we sample only 3,000 training data samples from the original data, the imbalance of the training data could become "better" for larger instances of Pancake and BW. For example, if we sample 3,000 data from BW-5 or Pancake-7, the data would likely include some of the outliers. However, if we sample from BW-13 or Pancake-16, the probability of selecting the outliers becomes smaller. My hypothesis is that for smaller size problems, the 3,000 data samples are more likely to have samples with very small ℎ∗ values. And if a model overfits the 3,000 data samples, the more inaccurate it becomes (in terms of overall accuracy). Do the authors think this hypothesis is reasonable? The results in Figure 3-7 are highly affected by these details of the experimental setup. Therefore, I have a concern that these results do not relate to the theoretical analysis provided in section 4. Could the authors elaborate more on this?
> - Answer: Your hypothesis is reasonable. That is, if the training set consists of a smaller label range then fitting should become “easier” i.e., require a smaller neural network.  However, for the reported domains, we observe that the range of h* values in the dataset increases with the instance size. For instance, in our dataset, athe 4 pancake problem has 4 unique h* labels from [1,4] in our dataset as opposed to athe 16 pancake problem that has 11 unique labels from [6,16]. Similarly, for 3 BW we see 4 unique labels from [1,4] for 3 BW whereas for 13 BW there are 19 unique labels from [1, 19]. This is because the addition of new (larger) h* values outweighs the reduction in the probability of sampling low h* values. To summarize, your hypothesis is reasonable but it does not seem to apply to the reportedour experiments.
> We agree that Figure 3-7 suggests overfitting after past some network size threshold. Moreover, we agree that this threshold is affected by the dataset size. We indeed observed that larger datasets cause the overfitting threshold to increase (yet similar trends are observed albeit shifted). However, these results do not contradict our theoretical claims. On the contrary, these trends follow what we expect to see following the theoretical claims.
> To entirely avoid this issue (overfitting past some point), we would need to train and test on the entire state space. However, it is impractical to collect the (s, h*(s)) pairs for the entire state space for such NP-H problems. To avoid confusion, we have already added text at the beginning of Section 5 where we clarify the goals of the experimental section, i.e., to demonstrate the impact of our theoretical claims for DNN-based estimators and fixed sized datasets as commonly used in literature (Agostinelli et al. 2019, Orseau & Lelis 2018, 2021). Specifically, see the text “The empirical study aims to investigate the minimal size of the deep neural networks (as universal function approximators) to fit a **fixed sized dataset** of (s, h*(s)) pairs with gradient-descent-based optimization methods. “

---

> > ### Comment · Reviewer_6bhh · 2023-10-16
> > **Response to the response**
> >
> > I agree with this part of the authors' comment: "However, these results do not contradict our theoretical claims."
> >
> > However, I am unconvinced with this part: "The experiments suggest that, under conventional practices, the exponential growth of parameters in deep neural networks is necessary to estimate the h* function with deep neural networks as the instance size increases."
> > What I see from the results is only the growing trends. I cannot distinguish polynomial growth and exponential growth from the results.
> >
> > It is often observed that only limited instances are difficult (e.g., needs exponential time to solve), and most instances are easy.
> > For example, in the well-known case of random 3-SAT, only the instances with certain clauses-variables ratio are difficult, and the rest are easy.
> > If a dataset consisting only of the easy instances is used for the training, it is doubtful that the trained model would behave as theoretically expected.
> > It is questionable whether the experimental results in Section 5 are strongly related to the theoretical results in Section 4 and rather influenced by how the training data were prepared.
> > The authors' responses did not solve this concern.

---

> > > ### Author Response · Authors · 2023-10-16
> > > **Authors Response to Reviewer 6bhh's concern**
> > >
> > > However, I am unconvinced with this part: "The experiments suggest that, under conventional practices, the exponential growth of parameters in deep neural networks is necessary to estimate the h* function with deep neural networks as the instance size increases." What I see from the results is only the growing trends. I cannot distinguish polynomial growth and exponential growth from the results.
> > >
> > > - Answer: We understand your concern that the quoted text might be an overstatement. In order to be more precise, we changed the text to “The experiments suggest that, under conventional practices, the log of the observed number of parameters (y-axis) grows quickly with increasing instance sizes in the reported range. This initial increase (pre-overfitting) in log scale can be explained by exponential growth (with some base of exponent) as anticipated by our theoretical results”. We reflected this change throughout the paper for accuracy.
> > >
> > > It is often observed that only limited instances are difficult (e.g., needs exponential time to solve), and most instances are easy. For example, in the well-known case of random 3-SAT, only the instances with certain clauses-variables ratio are difficult, and the rest are easy. If a dataset consisting only of the easy instances is used for the training, it is doubtful that the trained model would behave as theoretically expected. It is questionable whether the experimental results in Section 5 are strongly related to the theoretical results in Section 4 and rather influenced by how the training data were prepared. The authors' responses did not solve this concern.
> > >
> > > - Answer: We agree with the example provided by the reviewer, that is, some subset of an NP-H problem might be in P. Moreover, we agree that there is a non-zero probability that all of the sampled instances in a training set might belong to such a subset. In such cases, we would expect that fitting the training data would be easier (possibly not requiring an exponentially growing DNN), opposing our theoretical claims. While the results cannot be considered as a validation of the theoretical results, they suggest that fitting an accurate heuristic function to random fixed-sized datasets of NP-H problem instances requires a DNN  whose size grows quickly. These results are in line with our theoretical results (they do not contradict super polynomial growth). While our conclusion from the experiments has little impact on the theoretical claims, we believe they have practical implications on existing solutions from previous work that also use finite-sized datasets sampled by performing random walks  (Agostinelli et al. 2019, Orseau & Lelis 2021). Given the concern, we have emphasized this further while discussing the results (see Conclusion) to avoid misinterpretation.

---

### Author Response · Authors · 2023-09-28
**Thanks for valuable feedback**

We appreciate the reviewers’ efforts and overall positive assessment of our work. Please see our responses and let us know if you have any more questions.

---

### Decision · Action_Editor_mfHe · 2023-10-23

**Recommendation:** Accept with minor revision

**Comment:**

This work is reviewed by three reviewers with diverse expertise. The authors have provided thorough responses to address the diverse concerns raised by all reviewers. After two rounds of rebuttals, two reviewers (scU8 and oyZD) vote for a clear accept, and one reviewer (6bhh) leans toward reject.

According to comments in the official recommendation, reviewer 6bhh likes this work's theoretical analysis but finds the experimental results unconvincing. On the other hand, this reviewer also mentions they are leaning toward accept if the experiment section is omitted, agree with the author's comment "However, these results do not contradict our theoretical claims", and believe "supporting a theoretical claim in computational complexity with empirical results is generally very challenging".

I read the paper in detail and agree with all reviewers that this work is well written, the theoretical results are clear and convincing, and at least some individuals in the TMLR audience will find this work interesting. However, the concerns raised by reviewer 6bhh on the experimental study are also valid and valuable. Several claims made in the original experiment section are indeed overstated and should be carefully revised. Given the authors have already successfully addressed some of reviewer 6bhh's concerns and have provided a reasonable response to the issues left, I think the remaining concerns can be addressed via a minor revision, and hence recommend **accept with minor revision**.

In the revised paper, I expect the authors to

+ Carefully revise the experiments section (section 5) and make sure all the claims are solid and well-supported. In addition, make sure all discussions provided in the responses are well incorporated into the revised paper.

+ I suggest the authors clearly discuss all the assumptions and limitations for their experimental settings early in section 5 (e.g., a separate subsection might be appropriate). This explicit discussion can avoid any misleading information or misunderstanding from the reader.

In addition, I also have a question on the discussion of future research direction, which will not affect my decision:

+ In the conclusion section, the authors expect the research community should "(shift) its research efforts to other/additional ways of integrating heuristics search with machine learning" due to the unscalability of informed heuristic function estimation as discussed in this work. However, according to Bengio et al. [1], in practice, the learning-based approaches usually do not aim to solve the whole NP-Hard problem (e.g., all TSPs) but only care about a small subset of instances with a specific pattern (e.g., the pizza delivery problem in a small town, with daily different but similar patterns). In this case, will we still need an exponentially large model to learn a good model-based solver? This question is also related to the easy subset example provided by reviewer 6bhh and the NP-hard trade-off example provided by reviewer scU8.


[1] Yoshua Bengio, Andrea Lodi, Antoine Prouvost, Machine Learning for Combinatorial Optimization: a Methodological Tour d'Horizon, European Journal of Operations Research 2020.

**Audience:**

Building a learning-based solver to tackle NP-hard combinatorial optimization problems is an interesting and important research direction recently proposed in the research community. The generalization and (un)scalability is a critical issue for the learning-based solver. All reviewers believe this work is interesting to some individuals in TMLR's audience, and I also agree.

**Claims And Evidence:**

This work studies the (un)scalability issue for a recent line of research that builds deep neural network models to serve as a completely informed heuristic function for NP-hard search problems. It provides a theoretical analysis to show that, to learn a sufficiently accurate heuristic estimation for A* search, the model size must grow super-polynomial if $\text{NP} \not\subseteq \text{P/poly}$. This work also conducts various experiments with different models and loss functions to support the theoretical results.

After rebuttal, two reviewers believe the claims are well supported by the theoretical analyses and experiments. One reviewer likes the theoretical analyses but thinks the experimental results are not convincing.

---

> ### Author Response · Authors · 2023-11-18
> **Response to Action Editor**
>
> We would like to thank the reviewers and the action editor for their constructive comments and suggestions.
>
> 1. “Carefully revise the experiments section (section 5) and make sure all the claims are solid and well-supported. In addition, make sure all discussions provided in the responses are well incorporated into the revised paper.”
> - Answer: We have revised section 5 to make sure all claims are well supported. The changes were done as per our latest response to reviewer 6bhh. Specifically,
>
> a. We changed “grows exponentially” to “grows quickly” when discussing the trends.
>
> b. We also replaced “linear increase in log scale” with “grows quickly” while pointing out that the y-axis is on a log scale.
>
> c. We also added the following text when discussing the scalability trends “The trend shows that the minimum number of parameters grows quickly with the instance size. The observed trend can be explained by a super-polynomial growth (notice that the y-axis is in log scale), following our theoretical claims.”
>
> d. Changes (to text “grows exponentially/linear”) are also reflected in the abstract, introduction, and conclusion.
>
> 2. I suggest the authors clearly discuss all the assumptions and limitations for their experimental settings early in section 5 (e.g., a separate subsection might be appropriate). This explicit discussion can avoid any misleading information or misunderstanding from the reader.
> - Answer: Thanks for the suggestion. We have created a subsection to list the empirical limitations (Section 5.1).
> 3. In the conclusion section, the authors expect the research community should "(shift) its research efforts to other/additional ways of integrating heuristics search with machine learning" due to the unscalability of informed heuristic function estimation as discussed in this work. However, according to Bengio et al. [1], in practice, the learning-based approaches usually do not aim to solve the whole NP-Hard problem (e.g., all TSPs) but only care about a small subset of instances with a specific pattern (e.g., the pizza delivery problem in a small town, with daily different but similar patterns). In this case, will we still need an exponentially large model to learn a good model-based solver? This question is also related to the easy subset example provided by reviewer 6bhh and the NP-hard trade-off example provided by reviewer scU8.
> - Answer: We think the answer would depend on what is the subset being considered. If the subsets are NP-H (given a fixed subset selection strategy) and their decision variants can be represented by the k-cost-path problem, then we would still require a super-polynomially growing model to represent a sufficiently accurate heuristic estimator. This follows from our theoretical claims. If the subset is entirely in P, then the model size scaling is still an open question. If you consider a fixed-sized dataset as a subset selection strategy, as shown in the experiments, we do not see super-polynomial growth after a certain instance size. On a side note, the issue of scaling to larger problems is also briefly mentioned in the paper cited (Bengio et al. [1] Section 6.3 Scaling. Text “Indeed, all of the papers tackling traveling salesman problem through machine learning and attempting to solve larger instances see degrading performance as size increases much beyond the sizes seen during training”).

---

> > ### Comment · Action_Editors · 2023-11-19
> > **Minor revision for the link**
> >
> > Thank you for the camera-ready update. I've checked the updated manuscript and find everything works well except the broken link to the OpenReview page.
> >
> > Please fix the link to make it clickable and correctly link to the OpenReview page for this work.
> >
> > Congratulation and Best Regards,
> >
> > AE

---

> > > ### Author Response · Authors · 2023-11-19
> > > **Response to AE's comment**
> > >
> > > Thanks for a quick check.
> > >
> > > We have now verified the link to openreview works.